# Chronic ethanol consumption compromises neutrophil function in acute pulmonary *Aspergillus fumigatus* infection

Nathalia Luisa Sousa de Oliveira Malacco[1]*, Jessica Amanda Marques Souza[1], Flavia Rayssa Braga Martins[1], Milene Alvarenga Rachid[2], Janaina Aparecida Simplicio[3], Carlos Renato Tirapelli[3], Adriano de Paula Sabino[4], Celso Martins Queiroz-Junior[5], Grazielle Ribeiro Goes[6], Leda Quercia Vieira[7], Danielle Glória Souza[8], Vanessa Pinho[5], Mauro Martins Teixeira[7], Frederico Marianetti Soriani[1]*

[1]Department of Genetics, Ecology and Evolution, Universidade Federal de Minas Gerais, Belo Horizonte, Brazil; [2]Department of Pathology, Universidade Federal de Minas Gerais, Belo Horizonte, Brazil; [3]Department of Psychiatric Nursing and Human Sciences, Universidade de São Paulo, Ribeirão Preto, Brazil; [4]Department of Clinical and Toxicological Analysis, Universidade Federal de Minas Gerais, Belo Horizonte, Brazil; [5]Department of Morphology, Universidade Federal de Minas Gerais, Belo Horizonte, Brazil; [6]Department of Biology, Universidade do Estado de Minas Gerais, Divinópolis, Brazil; [7]Department of Biochemistry and Immunology, Universidade Federal de Minas Gerais, Belo Horizonte, Brazil; [8]Department of Microbiology, Universidade Federal de Minas Gerais, Belo Horizonte, Brazil

*For correspondence:
nathalialuisa2@gmail.com
(NLSOM);
fredsori@gmail.com (FMS)

**Competing interests:** The authors declare that no competing interests exist.

**Abstract** Chronic ethanol consumption is a leading cause of mortality worldwide, with higher risks to develop pulmonary infections, including *Aspergillus* infections. Mechanisms underlying increased susceptibility to infections are poorly understood. Chronic ethanol consumption induced increased mortality rates, higher *Aspergillus fumigatus* burden and reduced neutrophil recruitment into the airways. Intravital microscopy showed decrease in leukocyte adhesion and rolling after ethanol consumption. Moreover, downregulated neutrophil activation and increased levels of serum CXCL1 in ethanol-fed mice induced internalization of CXCR2 receptor in circulating neutrophils. Bone marrow-derived neutrophils from ethanol-fed mice showed lower fungal clearance and defective reactive oxygen species production. Taken together, results showed that ethanol affects activation, recruitment, phagocytosis and killing functions of neutrophils, causing susceptibility to pulmonary *A. fumigatus* infection. This study establishes a new paradigm in innate immune response in chronic ethanol consumers.

## Introduction

Ethanol abuse is a leading cause of mortality worldwide (*World Health Organisation, 2014*). Chronic alcohol consumption has been correlated, as a comorbidity, to a wide range of health conditions, including alcoholic liver diseases, cirrhosis and cancers (*Szabo and Saha, 2015*; *Pritchard and Nagy, 2005*; *Szabo, 1999*; *Bautista, 1999*; *Luján et al., 2010*). Moreover, alcohol abusers are prone to develop severe lung inflammatory and infectious diseases, including acute respiratory distress syndrome (ARDS) (*Liang et al., 2012*), pneumonia caused by *Streptococcus pneumoniae* (*Trevejo-Nunez et al., 2015*; *Tsuchimoto et al., 2015*; *Boé et al., 2001*), *Klebsiella pneumoniae* (*Yeligar et al., 2014*; *Ohama et al., 2015*), Respiratory Syncytial Virus (RSV) infection

**eLife digest** Alcoholism is a chronic disease that has many damaging effects on the body. Over long periods, excessive alcohol intake weakens the immune system, putting consumers at increased risk of getting lung infections such as pneumonia.

Some forms of pneumonia can be caused by the fungus *Aspergillus fumigatus*. This microbe does not tend to be a problem for healthy individuals, but it can be fatal for those with impaired immune systems. Here, Malacco et al. wanted to find out why excessive alcohol consumers are more prone to pneumonia.

To test this, the researchers used two groups of mice that were either fed plain water or water containing ethanol. After 12 weeks, both groups were infected with *Aspergillus fumigatus*. The results showed that alcohol-fed mice were more susceptible to the infection caused by strong inflammation of the lungs.

Normally, the immune system confronts a lung infection by activating a group of defense cells called neutrophils, which travel through the blood system to the infection site. Once in the right spot, neutrophils get to work by releasing toxins that kill the fungus. Malacco et al. discovered that after chronic alcohol consumption, neutrophils were less reactive to inflammatory signals and less likely to reach the lungs. They were also less effective in dealing with the infection. Neutrophil released fewer toxins and were thus less able to kill the microbial cells.

These findings demonstrate for the first time how alcohol can affect immune cells during infection and pave the way for new possibilities to prevent fatal lung infections in excessive alcohol consumers. A next step would be to identify how alcohol acts on other processes in the body and to find a way to modulate or even revert the changes it causes.

(*Meyerholz et al., 2008*) and aspergillosis (*Blum et al., 1978*; *Gustot et al., 2014*). The mechanisms associated with this increased risk of disease and death are poorly understood, however studies have suggested that certain aspects of immune function may be altered by chronic ethanol intake (*D'Souza El-Guindy et al., 2007*; *Lippai et al., 2013*; *Yen et al., 2017*; *Zhang et al., 2015*; *Gurung et al., 2009*).

A*spergillus fumigatus* is a ubiquitous and saprophytic fungus whose conidia are inhaled by humans on a daily basis (*Latgé, 1999*). Immunocompromised individuals are considered the risk group to develop the pulmonary invasive aspergillosis (IA) (*Latgé, 2001*) and mortality rates reach up to 95% (*Brown et al., 2012*). In normal conditions, inhaled conidia are cleared through mucociliary actions. However, if conidia pass through the initial barrier, alveolar macrophage (AM) phagocytosis takes place, resulting in a cascade of cytokine and chemokine release to recruit neutrophils to prevent fungal development (*Dagenais and Keller, 2009*; *Caffrey-Carr et al., 2017*). In all these circumstances, an altered leukocyte function may be a major risk factor for IA. Despite all advance in diagnosis and treatment, aspergillosis' morbidity and mortality remain very high. Mildly immunocompromised conditions such as diabetes mellitus, low-dose corticosteroid therapy and alcoholism has been considered as predisposing factors (*Blum et al., 1978*; *Baddley, 2011*; *Kousha et al., 2011*).

Neutrophils have been shown to be essential to control fungal and bacterial burden in the site of infection and avoid the spread of these microbes and consequently, survival of the host (*Gazendam et al., 2016*; *Romani, 2000*; *Robertson et al., 2008*; *Ng et al., 2019*). During a pathogen-triggered inflammatory response, neutrophils are the earliest immune effector cells recruited to a site of infection (*Kolaczkowska and Kubes, 2013*; *Sun et al., 2014*; *Rios-Santos et al., 2007*). Neutrophil migration starts with the tethering and rolling of these cells on endothelial cells, a process mediated by selectins and their carbohydrate ligands on neutrophils and endothelial venules (*Soethout et al., 2002*). These interactions, together with chemokines signals such as CXCL1 and CXCL2 lead neutrophils to a crawling state through the endothelial vase (*Kolaczkowska and Kubes, 2013*; *Phillipson and Kubes, 2011*). G protein-coupled receptor on rolling neutrophils binds to the chemokines and changes β2 integrins conformation on the leukocyte surface, allowing a high affinity interaction with endothelial cells (*Zemans et al., 2009*).

Here, we describe that chronic ethanol consumption facilitates pulmonary infection by both *A. fumigatus* in mice. Mechanistically, chronic ethanol consumption impairs the normal neutrophil

migration to the site of infection *via* release of high levels of circulating chemokine CXCL1 after infection, followed by downregulation of its receptor 2 (CXCR2). Additionally, ethanol consumption is responsible for an impaired neutrophil function characterized by less phagocytosis, killing, and oxidative burst leading to elevated lung pathology in mice and accentuated mortality rates after infection.

## Results

### Chronic ethanol consumption did not induce systemic inflammation

To assess the relevance of ethanol chronic consumption, we characterized several parameters during chronic ethanol consumption on host before the infection (*Figure 1A*). First, we checked weight change, food and liquid consumption during ethanol treatment. We found that there was no weight change in mice during the treatment with ethanol (*Figure 1B*). Moreover, food and liquid consumption were diminished in ethanol-treated mice compared with control group (*Figure 1C and D*). Second, blood ethanol levels in mice were about 200 mg of ethanol per deciliter of whole blood while control group ethanol level were not detectable, measured by gas chromatography after 12 weeks of ethanol treatment, (*Figure 1E*). In order to verify if ethanol consumption affected hematopoiesis we analyzed bone marrow precursors and blood cell counts. Mononuclear and neutrophil counts in peripheral blood did not show differences between groups (*Figure 1F*). Chronic ethanol consumption did not affect granulocyte progenitors in bone marrow (*Supplementary file 1*). In addition, no changes were observed in ALT levels in mice serum after the ethanol treatment compared to the control mice group (data not shown). Finally, serum levels of inflammatory mediators, such as TNF-α, IL-1β and CXCL1, have not been changed in ethanol-fed mice (*Figure 1G–I*).

### Chronic ethanol consumption increased lethality and impaired pulmonary fungal clearance after *A. fumigatus* infection

After ethanol treatment, mice were intranasally infected with *A. fumigatus* conidia (*Figure 2A*). Ethanol-treated mice showed an increased lethality compared to control mice group (*Figure 2B*). In addition, weight loss was significantly higher in ethanol treated mice from days 3 to 7 after infection (*Figure 2C*). These clinical signs were accompanied by higher pulmonary fungal burden in ethanol-fed mice, demonstrating an impaired fungal clearance. Moreover, we were also able to identify hyphae into the airways of infected ethanol-treated mice, showed by red arrows at 24 hr after infection compared to control group (*Figure 2D–F*). Altogether, these results demonstrate that chronic ethanol consumption increased susceptibility to pulmonary *A. fumigatus* infection along with an impaired ability to clear the pathogen.

### Ethanol consumption altered cytokine release in airways in mice after *A. fumigatus* infection

Next, we assessed cytokine and chemokine levels by ELISA after *A. fumigatus* infection to determine whether levels of inflammatory mediators in BALF supernatants were altered after ethanol consumption. We found no significant differences in the levels of neutrophil chemotactic mediator CXCL1 after *A. fumigatus* infection in BALF of ethanol-treated mice compared to control group (*Figure 3A*). Interestingly, another CXCR2 agonist and neutrophil chemotactic mediator, CXCL2 were increased at 24 hr of infection in BAL fluid of ethanol-treated mice compared to control mice (*Figure 3B*). Alcohol consumption declined IL-17 levels after 24 hr of infection (*Figure 3C*). We also observed no differences in TNF levels between control and ethanol-fed mice group (*Figure 3D*). Moreover, ethanol consumption was able to down modulate IL-1β and IL-10 levels after fungal infection (*Figure 3E and F*). These results suggest that chronic ethanol consumption dysregulate cytokine and chemokine release post *A. fumigatus* infection.

### Ethanol consumption impaired neutrophil and lymphocyte recruitment into the airways during *A. fumigatus* infection

To clarify whether the diminished fungal clearance is due to an impaired inflammatory response, we next observed cell influx into the site of *A. fumigatus* infection. Although both groups exhibited a large recruitment of total cells into the airways, ethanol-treated mice had a significant reduction of

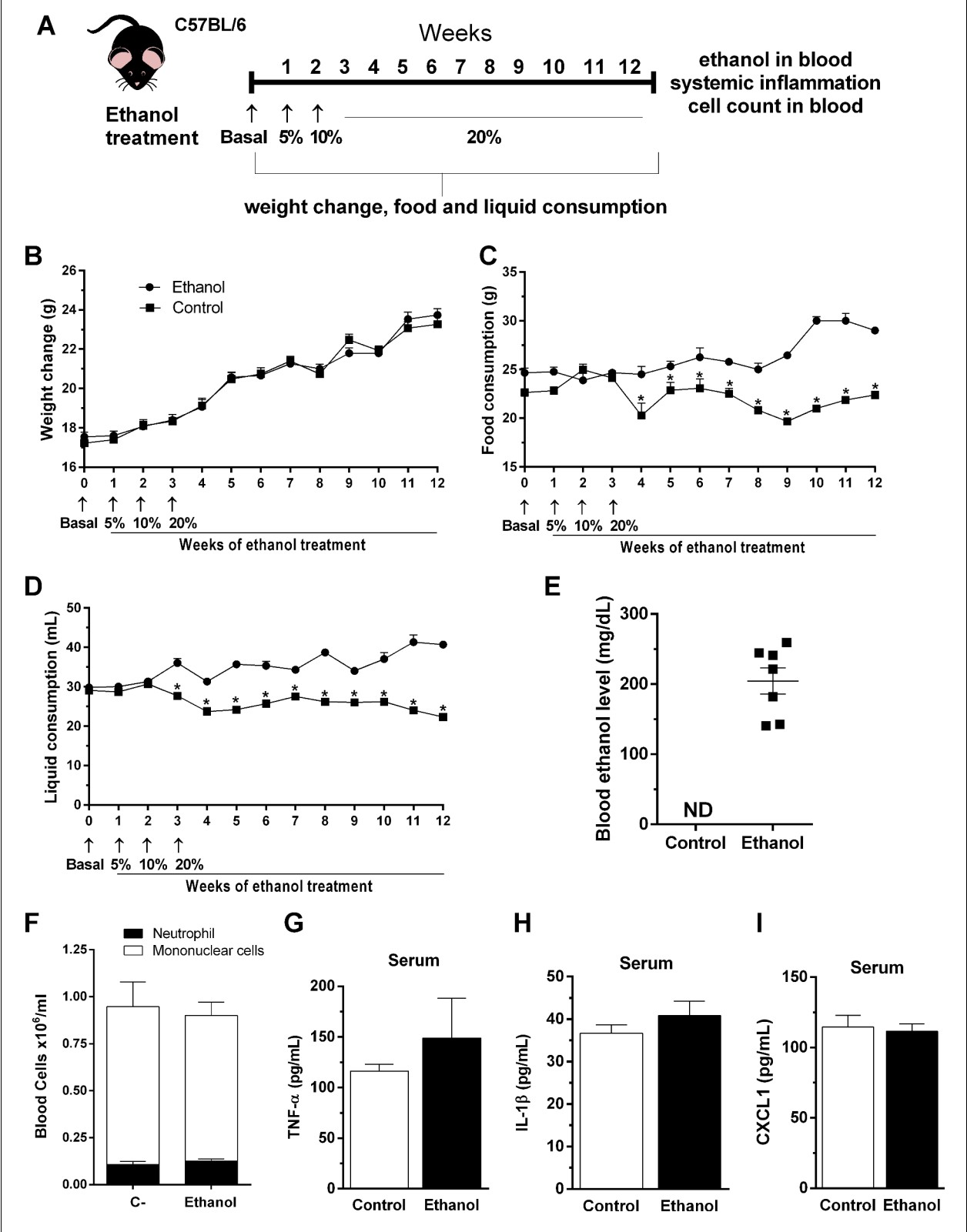

**Figure 1.** Outcome of chronic ethanol treatment in mice. (**A**) Experimental design: C57BL/6 mice received ethanol 5% (v/v) in the first week, followed by 10% (v/v) in the second week (to help mice acclimate with this intervention) and were treated during 10 weeks with ethanol 20% (v/v) in their drinking water. Control group received water. (**B–D**) During treatment, weight change, food and liquid consumption were measured. Data are presented as Mean ± SD (15 mice per group) *Significantly different ($p < 0.05$) in t test. After ethanol treatment, blood was collected to evaluate (**E**) blood ethanol

*Figure 1 continued on next page*

*Figure 1 continued*

levels, (F) differential cell count and (G) TNF-α, (H) IL-1β and (I) CXCL1 in serum. Data are presented as Mean ± SD (4 to 9 mice per group) and analyzed with ANOVA test. Please, also see *Supplementary file 1* and *Figure 1—source data 1*.

The online version of this article includes the following source data for figure 1:

**Source data 1.** Values for the outcome of ethanol treatment in mice.

recruited cells, compared to control group 24 hr after infection. However, at 48 hr after infection, there was a similar cell migration into the site of infection in both groups (*Figure 4A*). We demonstrate that almost all cells migrated to the airways represent neutrophils, one of the most important cells involved in killing and control of *A. fumigatus* infection (*Erwig and Gow, 2016*). Neutrophils and lymphocytes into airways were significantly decreased in ethanol-treated mice at 24 hr post infection (*Figure 4B and C*). In contrast, there were no differences in macrophages and eosinophils recruited to the site of infection (*Figure 4D and E*). Regarding to inflammatory cells recruited to lung tissue, we also observed a diminished neutrophils migration into the lung parenchyma in ethanol-treated mice, by MPO measurement, at 24 hr after infection (*Figure 4F*). Similar to alveoli, eosinophils and macrophages in pulmonary tissue exhibited no differences after *A. fumigatus* infection (*Figure 4G and H*). In addition, both ethanol-treated and control groups exhibited similar number of leukocytes in blood (*Figure 4I*). Besides, we characterized the populations of lymphocyte migrating to the airways of infected mice. We assessed $CD3^+CD4^+IL17^+$ cells and results demonstrate that chronic ethanol consumption strongly reduced lymphocytes and IL-17 production in the site of infection after 24 hr of infection (*Figure 4J–L*). These results indicate that chronic ethanol consumption mostly affected specifically neutrophils recruitment to airways after *A. fumigatus* infection.

## Chronic ethanol consumption increased lung pathology after *A. fumigatus* infection

We assessed histopathology to determine the effect of ethanol consumption in pulmonary tissue after infection. Tissue sections of infected mice revealed a massive leukocyte recruitment into the lungs at 24 hr after infection, in which the inflammatory infiltrate covers a large part of the pulmonary parenchyma structure, including alveoli and perivascular regions that decreases after 48 hr of infection in the control group. The cellular infiltrate tissue remains more prominent after 48 hr of infection only in ethanol-fed mice (*Figure 5A*). Histopathology score results showed that both ethanol-treated and control groups had similar levels of inflammatory infiltrate, interstitial and alveolar edema and hemorrhage scores at 24 hr of infection. However, after 48 hr of infection, ethanol consumption showed a remaining cellular infiltrate, higher edema and hemorrhage, which increased the total pathology scores (*Figure 5B–D*).

## Chronic ethanol consumption reduced leukocyte rolling, adhesion, and chemotaxis in mouse neutrophils

In order to investigate the transmigration process of neutrophils upon chemotaxis in vivo, we first performed an intravital microscopy to visualize migratory cells (*Figure 6—source data 1*) (*Figure 6-rich media videos*). Intracrostal administration of LPS was not able to induce a strong increase in rolling and adherence of cells to post capillary venules in ethanol-fed group after 2 hr of stimulation, compared to stimulated control group (*Figure 6A–D*). In addition, to confirm the impaired migratory ability of neutrophils was caused by ethanol consumption in mice, we performed ex vivo chemotaxis assay towards CXCL2, with mouse bone marrow-derived neutrophils (*Figure 6E*). As we expected because of previously results showed in *Figure 4B*, neutrophils from ethanol-treated mice had an impaired migration towards the chemokine stimuli compared to neutrophils from control non-treated mice (*Figure 6F*). Taken together, these results demonstrate that chronic ethanol ingestion can affect neutrophil rolling, adhesion and recruitment in different tissues.

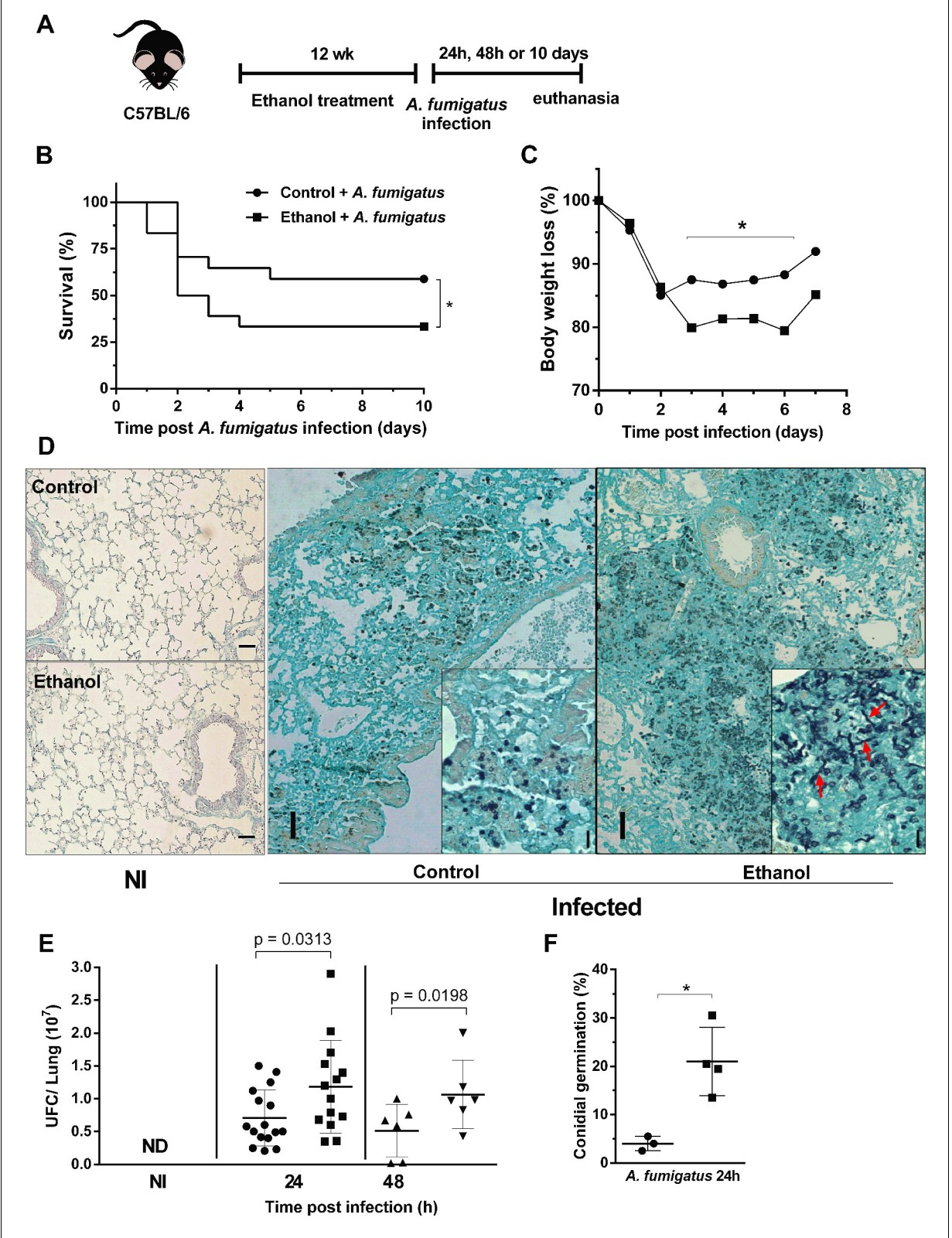

**Figure 2.** Chronic ethanol consumption leads to susceptibility associated with increased fungal load after *A. fumigatus* infection. (**A**) Experimental design: C57BL/6 mice were treated for 12 weeks with ethanol. The next day after the end of the treatment, mice were intranasally infected with $3 \times 10^8$ conidia of *A. fumigatus*. (**B**) Comparative lethality curves with P = <0.0001 in Log-rank (Mantel-Cox) test. (**C**) Comparative weight change curves of ethanol-fed group and control group were performed (18 mice per group). Right lungs were collected 24 and 48 hr after the infection. Homogenate

*Figure 2 continued on next page*

*Figure 2 continued*

from right lungs were plated and CFUs were quantified. Left lungs were fixed with formaldehyde 4% and embedded in paraffin. Sections were stained with GMS and the percentage of germlings was counted (6–16 mice per group) P value indicated in the figure in t test. (D) Representative slides of GMS staining. The insets in 24 hr images represent magnification to show germlings (red arrows) into lung tissue. (E) Fungal load and (F) fungal germination in lungs p=0.0389 in t test. Bars represent 100 µm.

### Chronic ethanol consumption impairs neutrophil's activation and recruitment by modulation of CD11b, CD62L and down regulation of CXCR2 in these cells during *A. fumigatus* infection

Neutrophils chemotaxis and recruitment is a complex process that requires leukocyte-endothelial interactions as well as inflammatory mediators' release (*Kolaczkowska and Kubes, 2013*). Moreover, the main regulator of neutrophil migration in acute inflammation is CXCR2. To further investigate whether the impaired neutrophil migration is due a deficient neutrophil activation status during infection, we accessed the expression of the markers CD11b, CD62L and CXCR2 in circulating

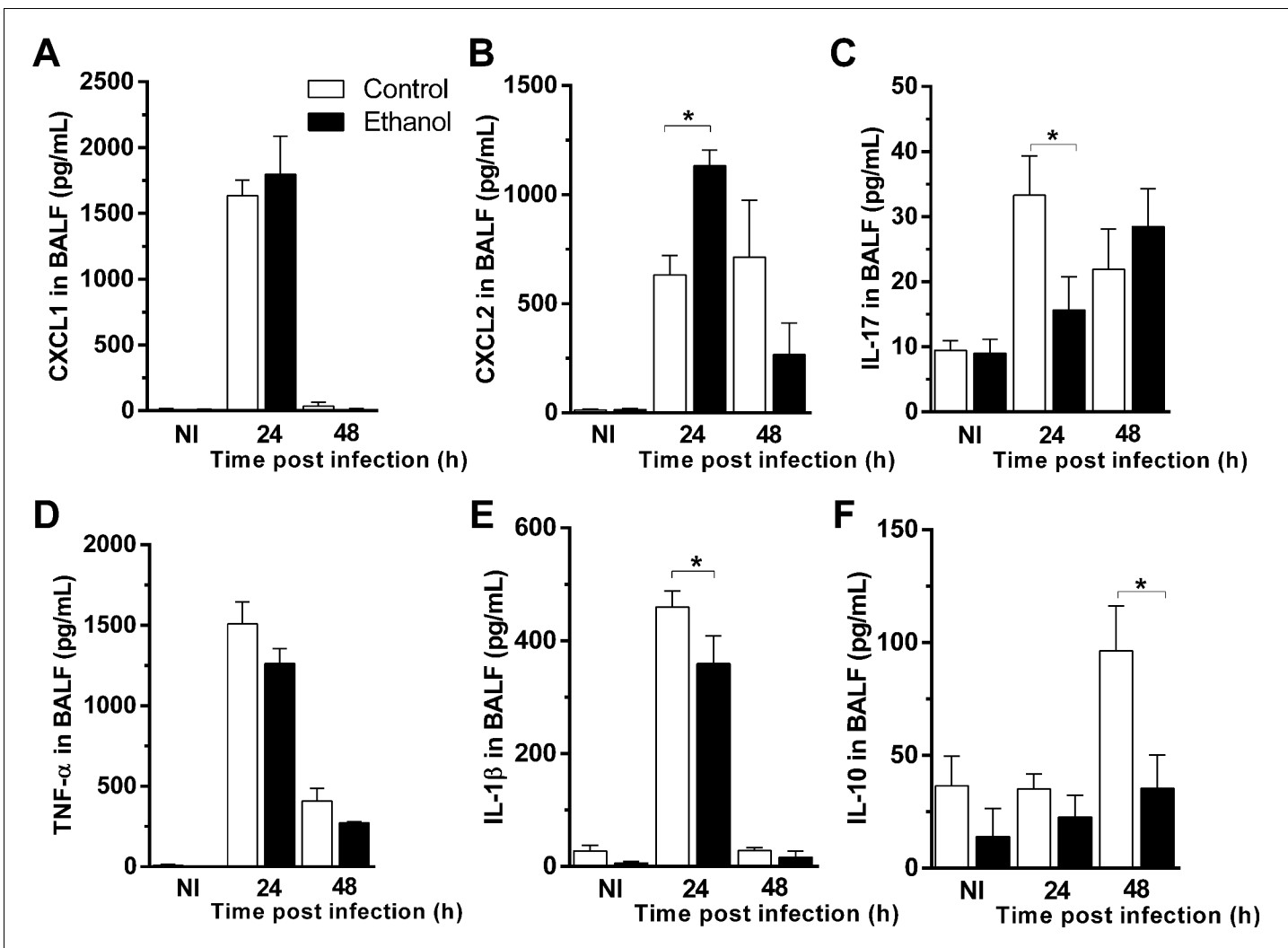

**Figure 3.** Ethanol consumption altered cytokine release in airways in mice after *A. fumigatus* infection. BALF supernatants were harvested at 24 and 48 hr after infection and used for ELISA assay. (A) CXCL1, (B) CXCL2, (C) IL-17, (D) TNF, (E) IL-1β and (F) IL-10 levels in BALF. Experiments were assayed in triplicate. Data are presented as Mean ± SD (3 to 9 mice per group). *p<0.0247 in ANOVA test. Please, also see *Figure 3—source data 1*.
The online version of this article includes the following source data for figure 3:

**Source data 1.** Values for inflammatory mediators in BALF after *A. fumigatus* infection.

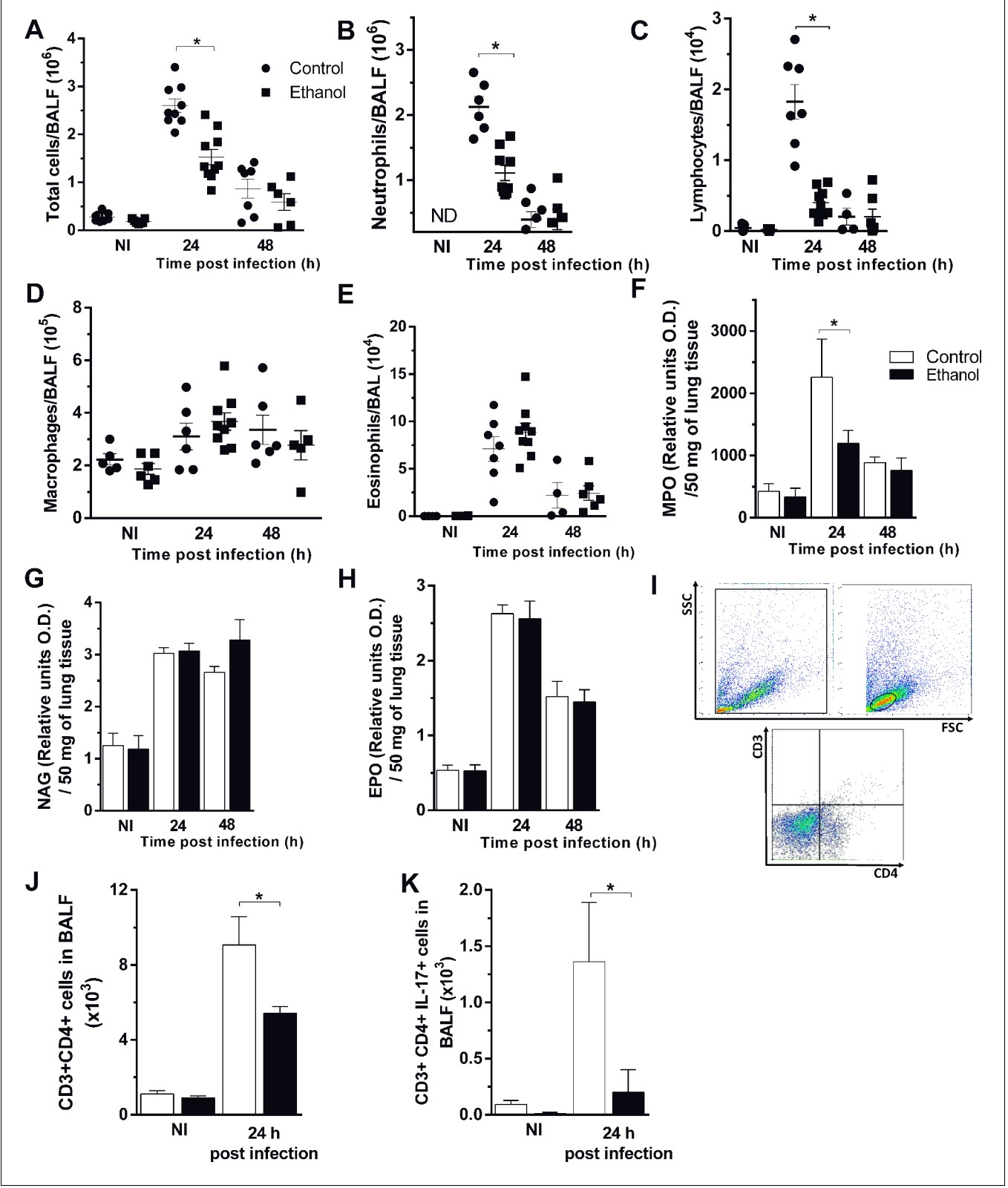

**Figure 4.** Chronic ethanol consumption affects cellular influx after *A. fumigatus* infection. After ethanol treatment, mice were infected with *A. fumigatus*. BALF and lungs were harvested at 24 and 48 hr after infection for inflammatory cell infiltrates determination. (**A**) Total cells, (**B**) Neutrophil, (**C**) Macrophage, (**D**) Lymphocyte, (**E**) Eosinophil counts in BALF. (**F**) MPO, (**G**) NAG and (**H**) EPO assays in lungs. (**I**) Blood leukocyte count/mL. BALF cells were labeled with specific antibodies for flow cytometry. (**J**) Gate strategy for CD3+CD4+ T cells. (**K**) CD3+CD4+ cells and (**L**) CD3+CD4+IL-17+ cells in

*Figure 4 continued on next page*

*Figure 4 continued*

BALF. Experiments were assayed in triplicate. Data are presented as mean ± SD (4 to 7 mice per group). *Significantly different (p<0.01) in ANOVA test. Please, also see *Figure 4—source data 1*.

The online version of this article includes the following source data for figure 4:

**Source data 1.** Values for indirect measurement of cell infiltration in lung tissue and CD3+CD4+IL-17 cells in BALF after *A. fumigatus* infection.

neutrophils after *A. fumigatus* infection using flow cytometry. While CD11b is mobilized from specific granules to the cell surface, enzymatic shedding in activated polymorphonuclear (PMN) rapidly down regulates CD62L. Both markers have their constitutive expression in resting PMNs. We found that neutrophils activation status was compromised, with CD62L up regulation and CD11b down regulation, in the peripheral blood neutrophils of infected ethanol-treated mice compared to infected non-treated mice (*Figure 7A–C*). In sepsis, the reduction of neutrophils migration is related to the down regulation of CXCR2 protein expression on circulating neutrophils surface (*Alves-Filho et al., 2009*). Indeed, CXCL1 serum levels were strongly augmented in ethanol-treated mice compared to the control group at 24 hr post infection (*Figure 7D*). We next analyzed the role of chronic ethanol intake in regulating CXCR2 expression in circulating neutrophils. At 24 hr after *A. fumigatus* infection, ethanol-treated mice exhibited significantly fewer circulating neutrophils expressing CXCR2 in the surface compared to the non-treated group (*Figure 7E and F*).

## Ethanol consumption decreases neutrophil functions of phagocytosis, killing and oxidative burst after *A. fumigatus* challenge

For the proper killing of *A. fumigatus*, fungal phagocytosis and ROS production by neutrophils are key events (*Philippe et al., 2003*). In this sense, we accessed phagocytosis and killing of *A. fumigatus* conidia by bone marrow-derived neutrophils from ethanol-treated and non-treated mice. We found that phagocytosis was significantly reduced in ethanol-treated mice either in vivo, evaluated by BALF recruited cells, or ex vivo, in bone marrow neutrophils, compared to control group (*Figure 8A and B*). Fungal killing was also reduced in ethanol intake mouse neutrophils (*Figure 8C*). Moreover, to evaluate the effect of chronic ethanol consumption in the promotion of respiratory burst of bone marrow-derived neutrophils we performed chemiluminescence experiments using luminol, which served as a probe for superoxide $(O_2^{\bullet-})$[49]. We observed that neutrophils from ethanol-fed mice produced lower levels of ROS triggered by *A. fumigatus* conidia compared to neutrophils from non-treated mice (*Figure 8D*). These results suggest that phagocytosis, killing and ROS production functions in neutrophils were affected by chronic ethanol consumption.

## Discussion

For several centuries, chronic ethanol consumption has been associated to increased susceptibility to infections as well as increased morbidity and mortality after injury (*Lieber, 2005*; *Molina et al., 2010*). Numerous studies have shown the effects of acute or chronic exposure to ethanol in inflammatory infections such as in models of *K. pneumoniae* infection and gut bacteria-associated sepsis (*Ohama et al., 2015*), intravenous *Escherichia coli* challenge (*Bagby et al., 1998*), *S. pneumoniae* (*Trevejo-Nunez et al., 2015*; *Boé et al., 2001*) and a few studies reporting this effect in *Aspergillus* infection (*Blum et al., 1978*).

In the present study, we demonstrate how chronic ethanol consumption alters immune and inflammatory pulmonary response after *A. fumigatus* infection. This involves several immunological phenomena, including defective leukocytes rolling and adhesion, impaired neutrophil migration by down regulation of CXCR2, failed neutrophil activation, impaired neutrophil effector functions (phagocytosis and killing), and reduction of polarized-Th17 innate response. Those features are responsible for an increased susceptibility of mice to *A. fumigatus* infection.

Chronic ethanol consumption did not alter the basal levels of inflammatory cytokines and chemokines before infection. Ethanol ingestion had an important role in modulating immune response after infection, which is in accordance with previous findings (*D'Souza El-Guindy et al., 2007*; *Bhatty et al., 2011*). Ethanol consumption also alters IL-6 and TNF expression of LPS-stimulated Kupffer cells (*Maraslioglu et al., 2014*). In the present work, we found reduction in IL-1β and IL-17 production after *A. fumigatus* infection in ethanol-fed mice, suggesting that the immunomodulatory

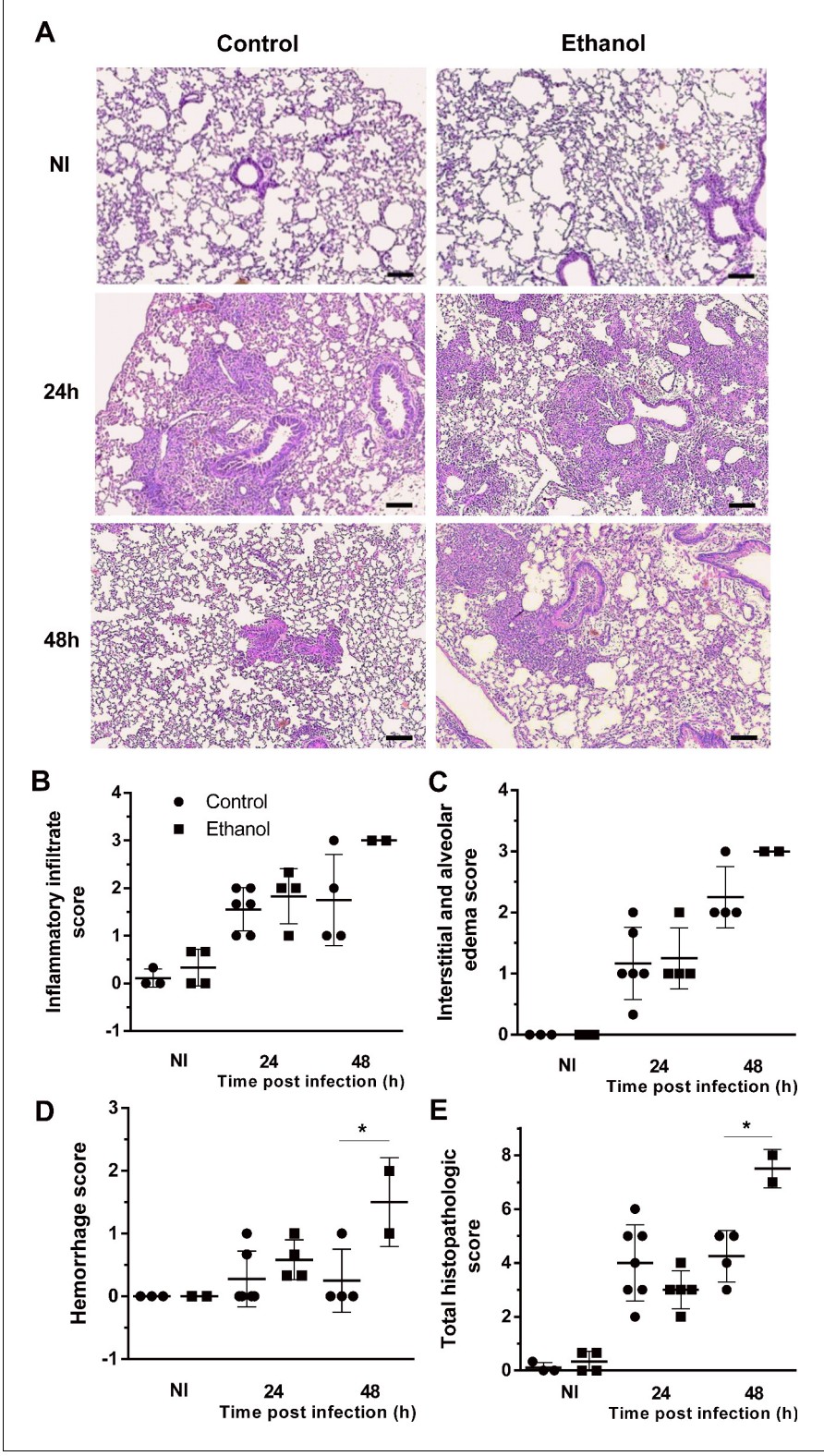

**Figure 5.** Lung histopathology is affected by ethanol treatment in mice. Lungs were harvested at 24 and 48 hr after infection fixed with formaldehyde 4% and embedded in paraffin. (**A**) Sections were stained with Hematoxylin and Eosin. Samples were graded on a 0 to 5-point scale score for (**B**) inflammatory infiltrate, (**C**) interstitial and alveolar edema, (**D**) hemorrhage and (**E**) total histopathologic score. Experiments were assayed in triplicate. Data are presented as mean ± SD (with 2 to 6 mice per group). *p<0.0142 in ANOVA test. Bars represent 100 μm.

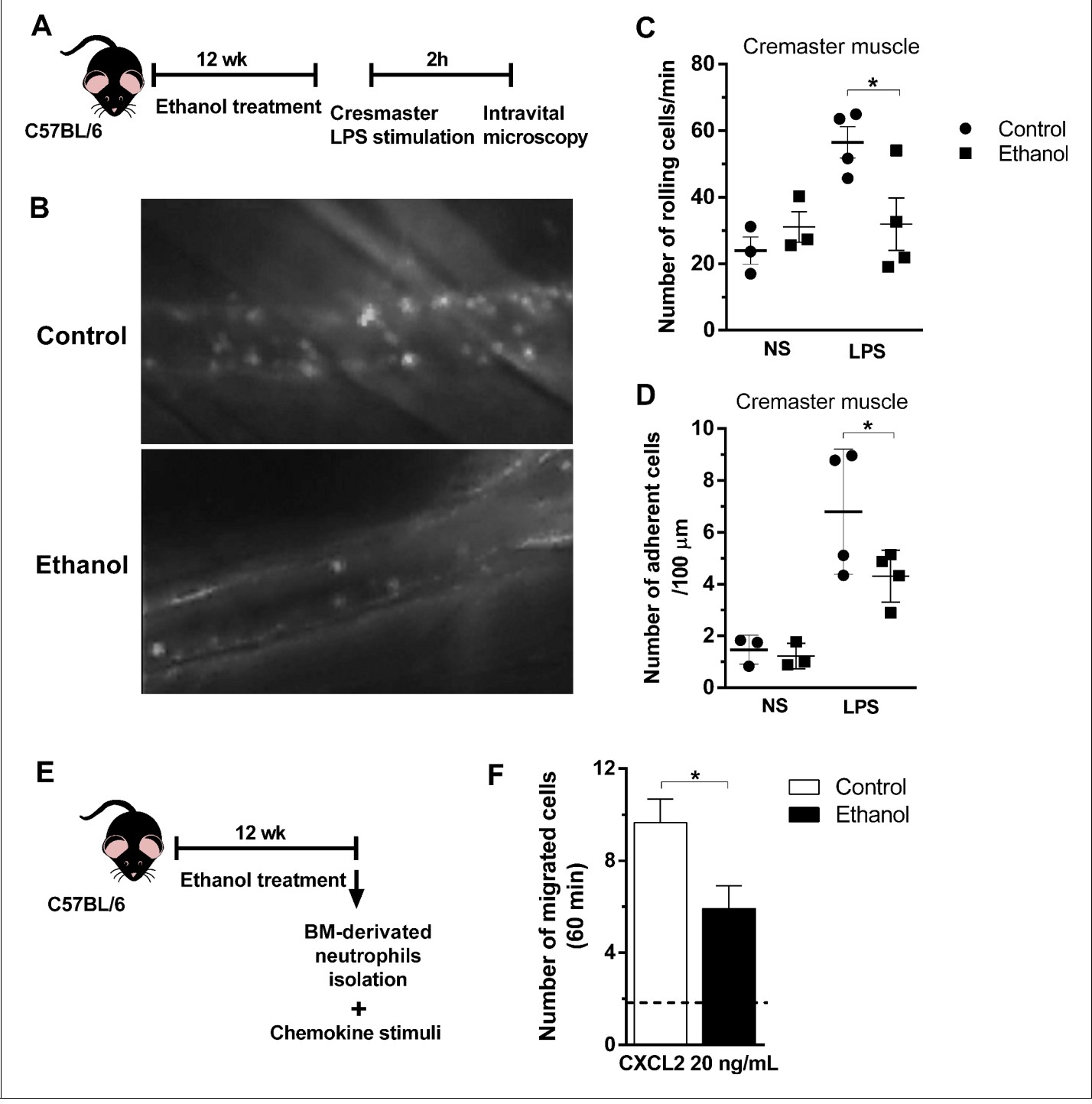

**Figure 6.** Chronic ethanol consumption reduced leukocyte rolling and adhesion in vivo and neutrophils chemotaxis ex vivo. (**A**) Experimental design: after the chronic ethanol treatment, mice received an intracrostal injection of LPS. After 2 hr, mice cremaster from mouse was exposed to examine the microcirculation by intravital microscopy. Post capillaries venules were recorded. (**B**) Representative images from the recorded videos (please see the rich media). (**C**) Number of rolling cells and (**D**) Number of adherent cells were counted in the videos. (**E**) Ex vivo neutrophil chemotaxis. Experimental design: after ethanol treatment, BM-derived neutrophils were separated by density gradient and a chemotaxis assay in a Boyden chamber towards CXCL2 was performed. (**F**) Number of migrated neutrophils in 60 min. Data are presented as Mean ± SD (3 to 5 mice per group) *Significantly different (p<0.0028) in ANOVA test. Please, also see *Figure 6—source data 1* and Figure 6-rich media videos.

The online version of this article includes the following video and source data for figure 6:

**Source data 1.** Values for bone marrow neutrophil chemotaxis.

*Figure 6 continued on next page*

*Figure 6 continued*
**Figure 6—video 1.** Leukocyte rolling and adhesion in vivo-control mice + vehicle.
https://elifesciences.org/articles/58855#fig6video1
**Figure 6—video 2.** Leukocyte rolling and adhesion*in vivo*-ethanol-treated mice + vehicle.
https://elifesciences.org/articles/58855#fig6video2
**Figure 6—video 3.** Leukocyte rolling and adhesion*in vivo*-control mice + LPS.
https://elifesciences.org/articles/58855#fig6video3
**Figure 6—video 4.** Leukocyte rolling and adhesion*in vivo*- ethanol-treated mice + LPS.
https://elifesciences.org/articles/58855#fig6video4

effects caused by chronic ethanol consumption may contribute to the impaired inflammatory response against *A. fumigatus*.

It has been demonstrated that chronic ethanol consumption causes cellular abnormality in mice lung resident cells. Alveolar macrophages from ethanol-fed mice had deregulation in NADPH oxidase system, impairing phagocytosis and killing against *K. pneumoniae* (*Yeligar et al., 2012*). In

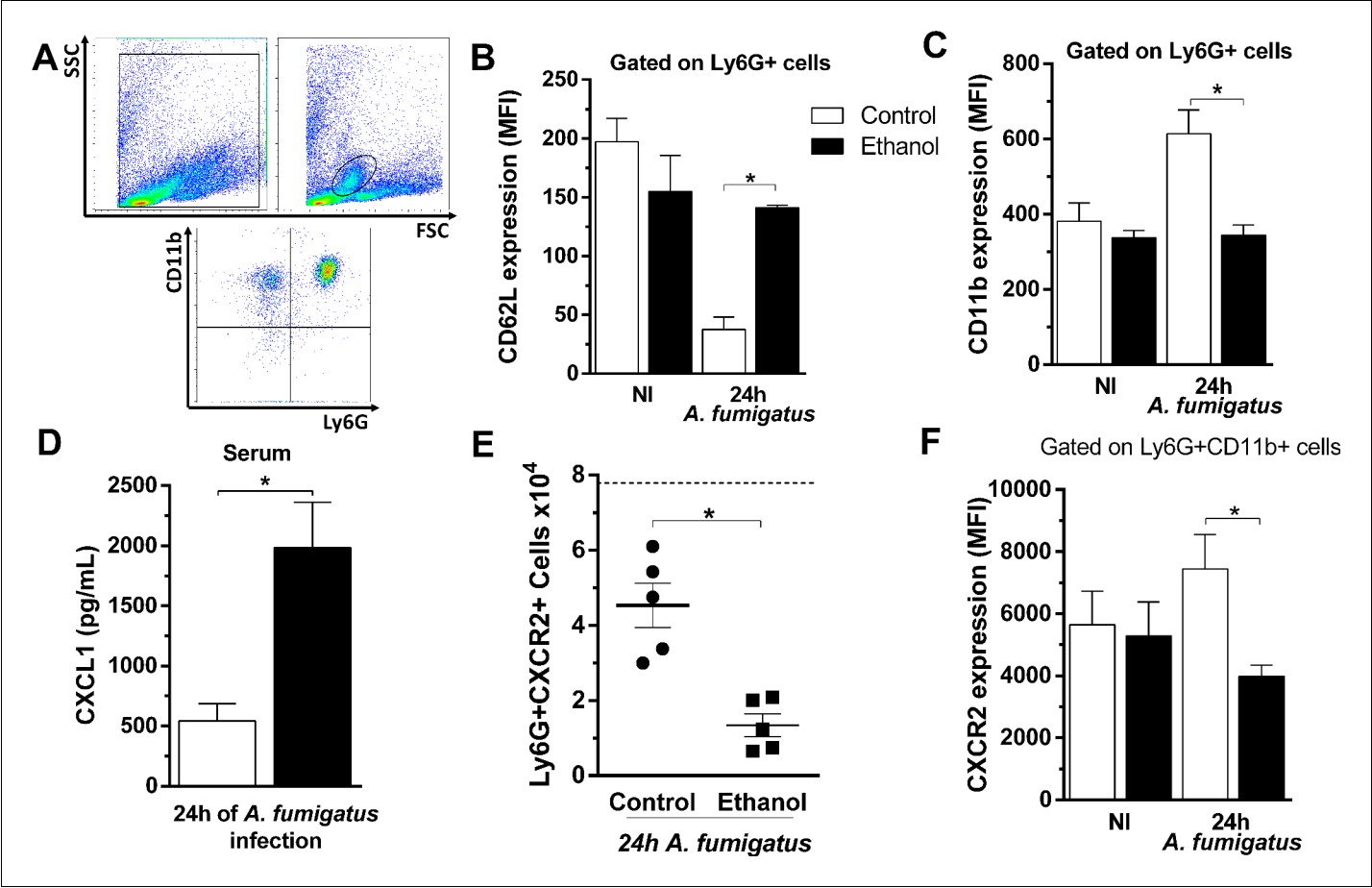

**Figure 7.** Chronic ethanol consumption impaired neutrophils activation and downregulated CXCR2 in a sepsis-like manner after *A. fumigatus* infection. After ethanol treatment and *A. fumigatus*, blood was harvested at 24 hr post infection. Neutrophils were labeled with specific antibodies for flow cytometry. (**A**) Gate strategy to analyze neutrophils. Neutrophils were gated by size and cellular complexity and then gated again as $Ly6G^+CD11b^+$ cells. (**B**) CD62L (*p=0.001) and (**C**) CD11b expression in circulating neutrophils (*p=0.0479). (**D**) Serum levels of CXCL1 were measured by ELISA assay after 24 hr of infection (*p = 0.0001). (**E**) $Ly6G^+CXCR2^+$ cells (*p=0.0489) and (**F**) MFI of CXCR2 expression in blood (*p=0.007). Experiments were done at least twice. Data are presented as mean ± SD (3 to 8 mice per group). Analysis were made by ANOVA test. Dashed line represents basal levels of non-infected groups. Please, also see *Figure 7—source data 1*.

The online version of this article includes the following source data for figure 7:

**Source data 1.** Values of flow cytometry data.

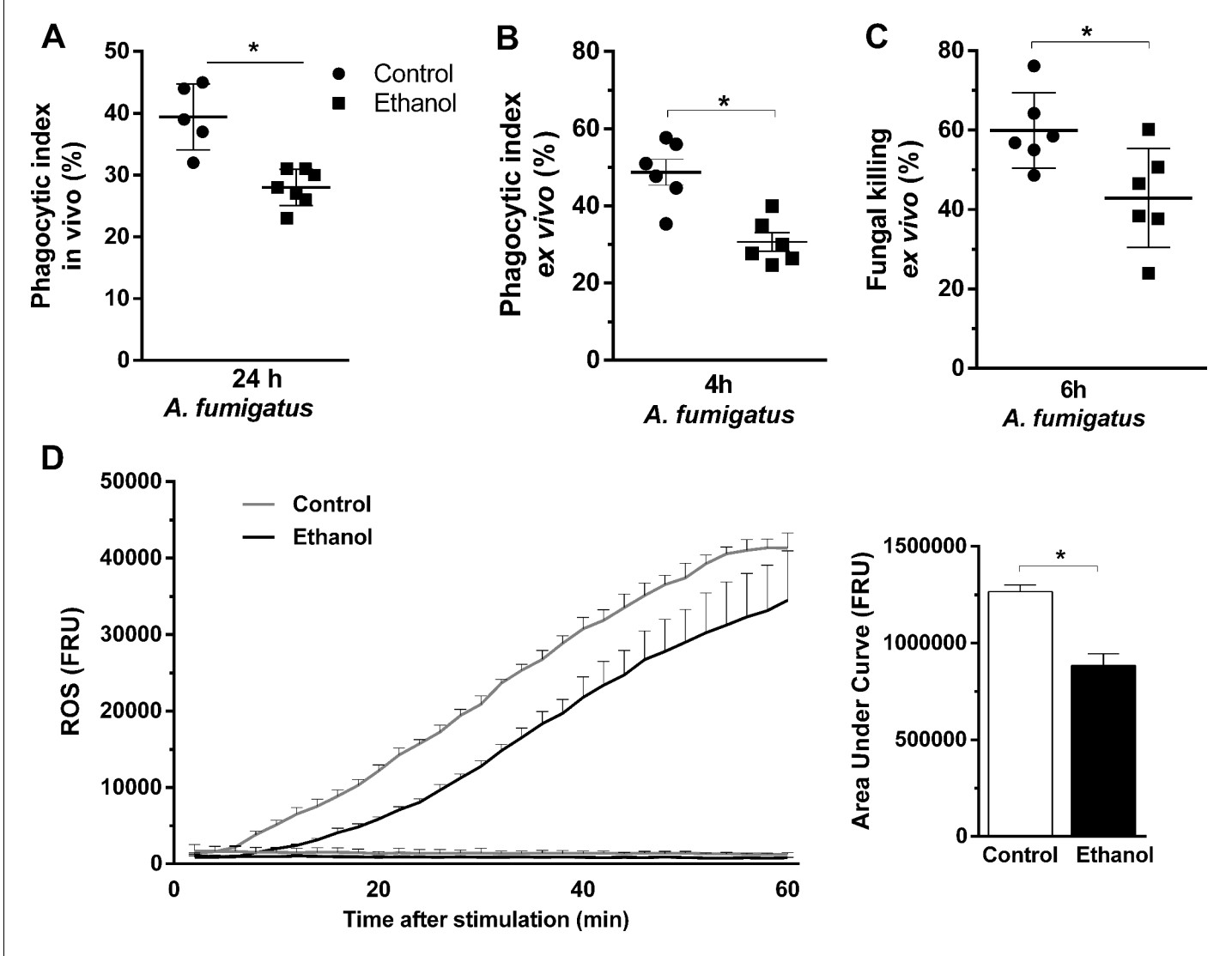

**Figure 8.** Ethanol consumption decreased neutrophils phagocytosis, killing and ROS production after *A. fumigatus* stimuli. (**A**) Conidial phagocytosis was determined in BALF (*p=0.007). (**B–D**) After ethanol treatment, BM-derived neutrophils were separated by density gradient and incubated with *A. fumigatus* conidia. (**B**) Ex vivo phagocytosis was assessed by cytospin preparations from BALFs (*p=0.0013). (**C**) Killing assay was evaluated by cell lysis with water, the diluted samples were plated in fungal medium and colony-forming units (CFU) were determined after overnight incubation (*p=0.0238). (**D**) Luminometry assay was performed to evaluate neutrophil-mediated ROS production and area under curve analysis. (*p=0.057). Data are presented as mean ± SD (3 to 6 mice per group). Analysis were made by t student test.

fact, an impaired NADPH oxidase activity is a well-known risk factor to develop invasive aspergillosis and other life threatening diseases, as seen in Chronic Granulomatous Disease (CGD) patients (*Cohen et al., 1981*; *King et al., 2016*; *Segal and Romani, 2009*). Another study showed both lymphocytes and neutrophils response impaired by ethanol consumption in a model of cutaneous infection by *Staphylococcus aureus*, in which chronic ethanol-fed mice showed great skin lesions and bacteremia associated with reduced IL-17 and IL-1β production, suggesting that Th17-mediated neutrophilic response was impaired (*Parlet et al., 2015*).

Neutrophil recruitment to the site of infection is essential for the control of invading extracellular pathogens (*Kolaczkowska and Kubes, 2013*; *Sun et al., 2014*; *Rios-Santos et al., 2007*). Neutrophils are major cell type recruited for *A. fumigatus* conidia and hyphae killing and neutropenic patients are more susceptible to systemic fungal infections (*Dagenais and Keller, 2009*; *Gazendam et al., 2016*). In our data, neutrophil recruitment to infection site, in ethanol-fed mice,

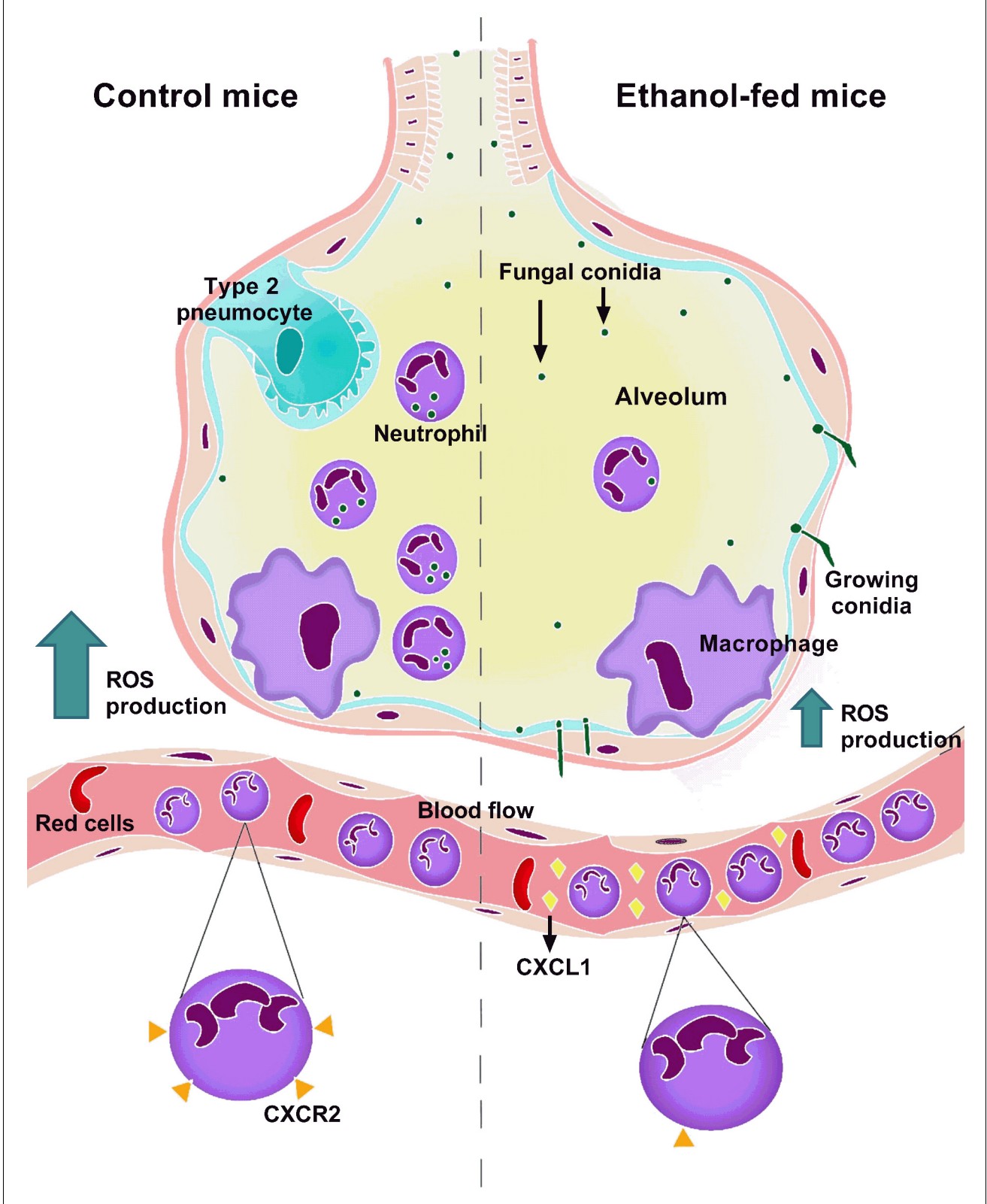

**Figure 9.** Mechanism related to chronic ethanol dysfunction in neutrophils after *A. fumigatus* infection. In normal conditions, infection with *A. fumigatus* in mice causes a huge inflammatory response, characterized by neutrophil chemokines release. The recruited neutrophils clear the infection by phagocytosis and ROS-mediated killing. In contrast, in a condition of chronic ethanol consumption, despite the correct induction of inflammatory

*Figure 9 continued on next page*

*Figure 9 continued*

response, there is an increase of CXCR2 ligands in blood flow, causing CXCR2 downregulation. This leads to lower neutrophils recruitment, culminating in substantial fungal burden into the lungs from ethanol-fed mice.

was impaired even though we did not observe decrease in both CXCL1 and CXCL2 BALF levels, indicating that ethanol consumption is responsible for compromising neutrophils activation and migration. Moreover, our data showed reduced leukocyte rolling and adhesion after LPS stimuli in ethanol-treated mice outside the airways. CXCL1 and CXCL2 chemokines bind to CXCR2 displaying an essential role in neutrophils activation and ensuing adhesion to endothelium (*Kolaczkowska and Kubes, 2013*; *Phillipson and Kubes, 2011*). In fact, it was demonstrated in vitro that chronic ethanol exposure impacted tight junction structures in epithelial cells, leading them vulnerable to endotoxemia (*Wood et al., 2013*). However, in our study vascular permeability was similar in both ethanol-fed mice and control mice groups (data not shown).

It has been well established that the decrease of CXCR2 expression impairs neutrophil migration (*Russo et al., 2009*), especially in sepsis (*Rios-Santos et al., 2007*; *Alves-Filho et al., 2009*), in which a great amount of inflammatory cytokines and chemokines is released to the blood, causing a complex systemic inflammation (*Pierrakos and Vincent, 2010*). In a model of severe sepsis, it was found that the migration failure and consequent mortality of individual was associated with the diminished expression of neutrophils CXCR2, which was due to great release of systemic CXCL1 (39). As seen in sepsis, our findings showed decrease CXCR2 expression and great amount of systemic CXCL1 levels in serum of ethanol-fed mice after *A. fumigatus* infection. To our knowledge, this is the first report establishing a relation between chronic ethanol intake and downregulation of CXCR2 receptor in mouse neutrophils. It is also important to mention that the cooperation between the GPCR receptor CXCR2 and P-selectin ligand, the P-selectin glycoprotein ligand-1 (PSGL-1) is essential to a successful neutrophil migration. The signaling events that ensure the adhesion cascade include the conversion of integrin $\alpha L\beta 2$ from a low-affinity conformation to an extended high-affinity conformation that causes arrest and, consequently, perivascular crawling (*Yago et al., 2018*). Further studies are needed to address the role of chronic ethanol consumption in expression of P-selectin and PSGL-1. Our data indicate that chronic ethanol consumption drives host to a sepsis-like phenotype and this mechanism is responsible for impaired neutrophil migration. The mechanisms whereby CXCR2 is downregulated need further investigation.

To conclude, the findings presented here indicate a new paradigm in how chronic ethanol consumption strongly impairs neutrophils host pulmonary defense against *A. fumigatus* infection. This infection in mice causes a great inflammatory response, with release of cytokines and chemokines that act in favor to recruit neutrophils into the alveoli and these neutrophils are able to clear the fungus. In contrast, in a condition of chronic ethanol consumption, despite the correct induction of inflammatory response, neutrophils exhibit failure in activation, through the down regulation of CD11b and up regulation of CD62L in blood neutrophils and by accentuated release of CXCR2 ligands in blood flow. This leads to CXCR2 down regulation, which culminated in impaired neutrophils recruitment, increased fungal load and exacerbated lung pathology in mice. We also associate the lower neutrophils levels into the airways with lower innate polarized-Th17 immune response and reduced phagocytosis and killing of *A. fumigatus* conidia. In fact, we observed growing conidia and substantial fungal burden in lung from ethanol-fed mice, contributing to the highest susceptibility of ethanol-treated mice to *A. fumigatus* infection (*Figure 9*).

## Materials and methods

### Key resources table

| Reagent type (species) or resource | Designation | Source or reference | Identifiers | Additional information |
|---|---|---|---|---|

*Continued on next page*

*Continued*

| Reagent type (species) or resource | Designation | Source or reference | Identifiers | Additional information |
|---|---|---|---|---|
| Strain, strain background (*Mus musculus*) | C57BL/6J | Multidisciplinary center for Biological Investigation on Laboratory Animal Science (CEMIB) -Unicamp | C57BL/6JUnib | |
| Genetic reagent (*Aspergillus fumigatus*) | Strain A1163 | *A. fumigatus* CEA17 isolate (CEA10 derivative) and converted to pyrG+ via *A. niger* pyrG gene ectopic insertion *Fedorova et al., 2008*; *Malacco et al., 2019*. | | |
| Cell line (*M. musculus*) | Primary bone marrow neutrophils | This paper | C57BL/6JUnib | Freshly isolated from C57BL/6J (*M. musculus*) |
| Peptide, recombinant protein | Recombinant murine MIP-2 (CXCL2) | PeproTech | Cat# 250–15 | Chemotaxis (20 ng/ml) |
| Antibody | Purified NA/LE CD16/CD32 Clone 2.4G2 - FC Block (Rat monoclonal) | BD Biosciences | Cat# 553140 | FACS (1:100) |
| Antibody | anti-CD3-FITC (Rat monoclonal) | BD Biosciences | Cat# 555274 | FACS (1:100) |
| Antibody | anti-CD4-APC (Rat monoclonal) | BD Biosciences | Cat# 553051 | FACS (1:200) |
| Antibody | anti-IL-17a-PE (Rat monoclonal) | BD Biosciences | Cat# 559502 | FACS (1:100) |
| Antibody | anti-Ly6G-BV421 (Rat monoclonal) | BD Biosciences | Cat# 562737 | FACS (1:50) |
| Antibody | anti-CXCR2-PE (Rat monoclonal) | R and D Systems | Cat# FAB2164P | FACS (1:10) |
| Antibody | anti-CD62L-APC (Rat monoclonal) | BD Biosciences | Cat# 553152 | FACS (1:100) |
| Antibody | anti-CD11b-FITC (Rat monoclonal) | BD Biosciences | Cat# 553310 | FACS (1:100) |
| Commercial assay or kit | Mouse TNF-a ELISA kit | R and D Systems | Cat# DY410 | |
| Commercial assay or kit | Mouse IL-1b ELISA kit | R and D Systems | Cat# DY401 | |
| Commercial assay or kit | Mouse CXCL1 ELISA kit | R and D Systems | Cat# DY453 | |
| Commercial assay or kit | Mouse CXCL2 ELISA kit | R and D Systems | Cat# DY452 | |
| Commercial assay or kit | Mouse IL-17 ELISA kit | R and D Systems | Cat# DY421 | |
| Commercial assay or kit | Mouse IL-10 ELISA kit | R and D Systems | Cat# DY417 | |
| Software, algorithm | Prism | GraphPad | | |
| Software, algorithm | FlowJo | BD | | |

## Ethics statement and mouse model of chronic ethanol consumption

All animal experiments received prior approval from the Animal Ethics Committee (CEUA) of Universidade Federal de Minas Gerais (UFMG), Brazil (Protocol number: 4/2015). Male C57BL/6J mice

were randomly allocated into experimental groups. Mice were maintained in specific pathogen-free conditions. The chronic ethanol consumption model used was previously described (*Ceron et al., 2018*; *Simplicio et al., 2017*). Briefly, mice received ethanol 5% (v/v) in the first week, followed by 10% (v/v) in the second week (to help mice acclimate with this intervention) and were treated during 10 weeks with 20% (v/v) of ethanol in their drinking water. Control group received water. This protocol replicates blood ethanol levels after chronic ethanol consumption in human subjects, as described in studies with C57BL/6J or BALB/c mice (*Ceron et al., 2018*; *Simplicio et al., 2017*; *Urso et al., 1981*). Weight change, food and liquid consumption were measured weekly during ethanol treatment.

## Cytokine and chemokine measurement

Cytokine and chemokine levels (TNF-α, IL-1β, CXCL1, CXCL2, IL-17, IL-10) were quantified in BAL, serum or plasma fluid using DuoSet ELISA kits (R and D Systems), in accordance to the manufacturer's instructions.

## Determination of blood ethanol levels

The determination of blood ethanol levels was made as previously described (*Gonzaga et al., 2015*). Briefly, mice were anesthetized and 100 μL of blood was collected and transferred into headspace vials. Blood ethanol levels were measured by gas chromatograph using a gas chromatographer as previously described (*Ceron et al., 2018*).

## Mice infection

To determine the impact of chronic alcohol intake in fungal pulmonary infection, male C57BL/6J mice were treated for 12 weeks. After the last day of treatment, mice were infected intranasally with *Aspergillus fumigatus* A1163 strain (*Fedorova et al., 2008*). The fungus was grown in complete media for 48 hr at 37°C (*Malacco et al., 2019*). Fungal conidia were harvested by washing the media with sterile phosphate-buffered saline (PBS). After filtering, conidia were centrifuged at 1400 x g resuspended and counted in Neubauer chamber. Mice were infected with $3 \times 10^8$ conidia/animal, prepared in PBS.

## BAL and tissue extraction

After 24 or 48 hr of infection, mice were anesthetized with ketamine (100 mg/kg) and xilazine (6 mg/Kg) and blood smear and serum or plasma were collected. After that, mice were euthanized and bronchoalveolar lavage fluid (BALF) was harvested as previously described (*Malacco et al., 2019*). BALF total cell counts were determined by counting leukocytes in Neubauer chamber. Differential cell count and in vivo phagocytosis count were obtained from cytospin preparations (Shandon III). BALF supernatants were used for cytokines, chemokines and total protein measurements. Protein amounts were quantified in BALF samples using the Bradford assay (*Bradford, 1976*).

## Lung pathology analysis

At the indicated time points, lungs were collected. The right lobes were removed and frozen for subsequent analysis of *myeloperoxidase* (MPO) (*Huang et al., 2016*), *N-acetyglucosaminidase* (NAG) (*Reiner et al., 1981*), *eosinophil peroxidase* (EPO) (*Strath et al., 1985*) or measurement of fungal burden. The left lobes were fixed in formalin 4% (v/v) for histopathological analysis. Formalin-fixed tissue was dehydrated gradually in ethanol, embedded in paraffin, and 4 μm sections were stained with Hematoxylin and Eosin (H and E) or Grocott's methenamine silver (GMS). The total histopathology score considered inflammatory infiltrate, interstitial and alveolar edema and hemorrhage (*Hubbs et al., 1997*). The percentage of germination of *A. fumigatus* conidia was counted in 200 to 300 fungal conidia in GMS-stained slides at x 100 magnification microscope.

## Flow cytometry

Leukocytes obtained from BAL or blood samples were subjected to hypotonic lysis to remove residual erythrocytes, as described previously (*Russo et al., 2009*). Briefly, cells were treated with Fc block (R and D Systems), labeled with relevant antibodies, namely: CD3 - fluorescein isothiocyanate (FITC), CD4 - APC, IL-17 - phycoerythrin (PE), Ly6G - brilliant violet 421 (BV421), CXCR2 - PE, CD62L

– APC and CD11b – FITC or isotype control. At least 30,000 events were acquired in a FACScan cytometer, and data were analyzed using FlowJo (Tree Star, Ashland, OR, USA) software. The relevant populations were gated, using accepted criteria for granularity, and sized and evaluated for staining of relevant surface and intracellular markers.

## Intravital microscopy

The mouse cremaster preparation was used to study the behavior of leukocytes in the microcirculation and adjacent connective tissue, as previously described (*Pinho et al., 2007*). Briefly, 2 hr prior the surgery, mouse cremaster muscle were injected with LPS (250 ng/mL) diluted in saline. Then an incision was made in the scrotal skin to expose the cremaster muscle, which was then carefully removed from the associated fascia. A lengthwise incision was made on the ventral surface of the cremaster muscle using a cautery. The testicle and the epididymis were separated from the underlying muscle and were moved into the abdominal cavity. The muscle was then spread out over an optically clear viewing pedestal and was secured along the edges with a 4–0 suture. The exposed tissue was superfused with warm PBS. An intravital microscope (Olympus BX50F4) with x 20 magnification objective lens and x 10 times magnification eyepiece was used to examine the cremasteric microcirculation. A video camera (5100 HS; Panasonic) was used to project the images onto a monitor, and the images were recorded for playback analysis. The numbers of rolling and adherent leukocytes were determined offline during the video playback analyses. Leukocytes were considered adherent to the venular endothelium if they remained stationary for at least 30 s. Rolling leukocytes were defined as white cells moving at a velocity slower than that of the erythrocytes within a given vessel.

## BM-derived neutrophils, phagocytosis and killing of fungal conidia

After isolation of mouse bone marrow (BM) from femur and tibia in RPMI 1640 medium, neutrophils were separated by density gradient centrifugation using Histopaque 1077 (density, 1.077 g/ml) in a 15 ml conical tube. Then, erythrocytes were lysed using ACK lysing buffer and the neutrophils were counted. Neutrophils purity was over than 80% (*Swamydas and Lionakis, 2013*). Phagocytosis and killing assay were performed by incubation of BM-derived neutrophils from mice treated and non-treated with ethanol with *A. fumigatus* conidia for 4 hr (phagocytosis) or 6 hr (killing) at 37° with 5% of $CO_2$ in the ratio of 5:1. Phagocytosis was evaluated in cytospin preparations. To determine *A. fumigatus* killing, cells were lysed with distilled water and the diluted samples were plated in fungal medium and colony-forming units (CFU) were determined after overnight incubation at 37°C, and the percentage of killing was calculated as a percentage of the viability after incubation without neutrophils.

## ROS detection

Luminometry assays were performed to evaluate the production of ROS by BM-derived neutrophils (*Goes et al., 2016*). Neutrophils ($1 \times 10^6$ cells/well) were resuspended in complete RPMI medium without phenol red. Then the cells were plated in 96 well opaque plates (NUNC, Rochester, NY, USA) with 0.05 mM luminol (5-Amino-2,3-dihydro-1,4-phthalazinedione; Sigma-Aldrich) and *A. fumigatus* conidia in the proportion of 10 conidia to one neutrophil. Measurements were taken for 60 min with 2 min interval. Production of ROS was assayed by the light intensity generated by the reaction between ROS and luminol and expressed as fluorescent relative units.

## Ex vivo chemotaxis assay

A modified Boyden chamber assay to examine the neutrophil chemoattractant response to CXCL-2 (kindly provided by Dr. José Carlos Alves-Filho, Universidade de São Paulo, Ribeirão Preto, Brazil) was performed using a 48-well microchamber (Neuro Probe) as previously described (*Pinho et al., 2007*). Murine bone marrow neutrophils were isolated as above described and resuspended in RPMI. Recombinant mouse CXCL-2 (20 ng/mL) diluted in running buffer (for wells containing neutrophils) or appropriate buffer control was added to the lower chambers of the apparatus. A 5-µm-pore polycarbonate membrane (Neuro Probe) was placed between the upper and lower chambers, and 5 $\times 10^4$ cells in a volume of 50 µL were added to the top chambers of the apparatus. Cells were allowed to migrate into the membrane for 1 hr per treatment at 37°C with 5% $CO_2$. Following

incubation, the chamber was disassembled and the membrane was scraped and washed three times in PBS to remove nonadherent cells before being fixed in methanol and stained using the Diff-Quik system (Dade Behring). Each well-associated membrane area was scored using light microscopy to count the intact cells present in five random fields.

## Statistical analysis

All experiments were made at least twice (biological replication) by independent experiments. The sample size estimation was done with G*Power 3.1 Software (Jacob Cohen's A power primer, 1992 in Psychological Bulletin Journal). Statistical analysis was performed with Graph Pad Prism six software (Graph Pad Prism Software, Inc, Sandiego, CA). Data are presented as the mean ± SD and were analyzed using One-way analysis of variance (ANOVA) followed by Tukey post-test to compare different groups. Student's t test was used to compare two groups. Survival analysis was made by Log Rank test. Statistical significance was set as $< 0.05$. For more information, please see the transparent reporting form.

## Acknowledgements

We would like to thank Universidade Federal de Minas Gerais for the opportunity to develop this work. We are thankful to Ilma Marçal and Rosemeire Oliveira for technical support.

# Additional information

### Funding

| Funder | Grant reference number | Author |
|---|---|---|
| Fundação de Amparo à Pesquisa do Estado de Minas Gerais | APQ-01756-10 | Jessica Amanda Marques Souza<br>Leda Quercia Vieira<br>Danielle Glória Souza<br>Vanessa Pinho<br>Mauro Martins Teixeira<br>Frederico Marianetti Soriani |
| Coordenação de Aperfeiçoamento de Pessoal de Nível Superior | 001 | Nathalia Luisa Sousa de Oliveira Malacco<br>Jessica Amanda Marques Souza |
| Conselho Nacional de Desenvolvimento Científico e Tecnológico | 474528-2012- 0 and 483184-2011-0 | Frederico Marianetti Soriani |
| Instituto Nacional de Ciência e Tecnologia em Dengue e Interações Microrganismo-Hospedeiro | | Nathalia Luisa Sousa de Oliveira Malacco<br>Jessica Amanda Marques Souza<br>Flavia Rayssa Braga Martins<br>Leda Quercia Vieira<br>Danielle Glória Souza<br>Vanessa Pinho<br>Mauro Martins Teixeira<br>Frederico Marianetti Soriani |
| Universidade Federal de Minas Gerais | 001 | Nathalia Luisa Sousa de Oliveira Malacco<br>Jessica Amanda Marques Souza<br>Flavia Rayssa Braga Martins<br>Milene Alvarenga Rachid<br>Celso Martins Queiroz-Junior<br>Grazielle Ribeiro Goes<br>Leda Quercia Vieira<br>Danielle Glória Souza<br>Vanessa Pinho<br>Mauro Martins Teixeira<br>Frederico Marianetti Soriani |

| Fundação de Amparo à Pesquisa do Estado de Minas Gerais | APQ-02198-14 and APQ-03950-17 | Jessica Amanda Marques Souza<br>Leda Quercia Vieira<br>Danielle Glória Souza<br>Vanessa Pinho<br>Mauro Martins Teixeira<br>Frederico Marianetti Soriani |
|---|---|---|

The funders had no role in study design, data collection and interpretation, or the decision to submit the work for publication.

## Author contributions

Nathalia Luisa Sousa de Oliveira Malacco, Investigation, Methodology, Writing - original draft; Jessica Amanda Marques Souza, Flavia Rayssa Braga Martins, Janaina Aparecida Simplicio, Celso Martins Queiroz-Junior, Grazielle Ribeiro Goes, Methodology; Milene Alvarenga Rachid, Carlos Renato Tirapelli, Vanessa Pinho, Resources, Methodology; Adriano de Paula Sabino, Formal analysis; Leda Quercia Vieira, Danielle Glória Souza, Resources; Mauro Martins Teixeira, Resources, Writing - review and editing; Frederico Marianetti Soriani, Conceptualization, Resources, Supervision, Funding acquisition, Project administration, Writing - review and editing

## Author ORCIDs

Frederico Marianetti Soriani (ID) https://orcid.org/0000-0003-4720-6746

## Ethics

Animal experimentation: This study was performed in strict accordance with the recommendations of the CONCEA (Conselho Nacional de Controle de Experimentação Animal) from Brazil. All animal experiments received prior approval from the Animal Ethics Committee (CEUA) of Universidade Federal de Minas Gerais (UFMG), Brazil (Protocol number: 4/2015).

## Decision letter and Author response

Decision letter https://doi.org/10.7554/eLife.58855.sa1
Author response https://doi.org/10.7554/eLife.58855.sa2

# Additional files

## Supplementary files

• Supplementary file 1. Percentage of neutrophil precursors in bone marrow after ethanol treatment.

## Data availability

All data generated or analysed during this study are included in the manuscript and supporting files.

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
