## [Decision Letter]

Thank you for submitting your article "Chronic ethanol consumption compromises neutrophil function in acute pulmonary Aspergillus and Streptococcus infections" for consideration by *eLife*. Your article has been reviewed by three peer reviewers, including Frank L van de Veerdonk as the Reviewing Editor and Reviewer #1, and the evaluation has been overseen by Jos van der Meer as the Senior Editor. The following individuals involved in review of your submission have agreed to reveal their identity: Agostinho Carvalho (Reviewer #2); Mark Gresnigt (Reviewer #3).

The reviewers have discussed the reviews with one another and the Reviewing Editor has drafted this decision to help you prepare a revised submission.

Summary:

Malacco and colleagues addressed the role of chronic alcohol consumption in the immune response of neutrophils to infection with the fungal pathogen Aspergillus fumigatus. The results were to an extent validated in models of bacterial infection. The authors reveal an increased susceptibility of ethanol-treated mice to fungal infection as the result of an impaired neutrophil recruitment and activation at the site of infection. Neutrophils from treated mice also displayed a general defect of the antifungal effector functions, including phagocytosis, and killing, which contributed to the susceptibility phenotype. In general, the manuscript presents interesting and provocative findings, but there are several issues to be clarified

Essential revisions:

1) The rationale to study aspergillosis in relation to chronic alcohol consumption is poorly justified in the paper. Indeed the authors cite the literature that ARDS is more common in chronic alcohol abuse and list a number of pathogens, but not Aspergillus. The rationale to study aspergillosis specifically remains unclear.

2) The use of an infection model with Streptococcal is used to validate the findings obtained with the aspergillosis model. However, it is not clear what is the actual relevance of adding this information, particularly in light of a few conflicting findings between the models, e.g., the levels of CXCL1 in the BAL. I would therefore recommend streamlining the manuscript to focus on the relevant findings on Aspergillus and removing the information on the Staphylococcus infection as it does not strengthen the main message of the manuscript.

3) The in vivo experiments look very rigorous, however, additional information should be provided before infection to see how ethanol-fed mice are affected by the pre-treatment period alone. This is information can also be vital to mechanistically understand the observed neutrophil defects

• Do the mice have systemic inflammation (elevated CRP/IL-1/TNF/CXCL1 levels)?

• Can endotoxin be detected systemically?

• Do the mice lose bodyweight as a result of chronic alcohol consumption?

• Are their differences in the absolute numbers of circulating immune cells (neutrophils/monocytes/lymphocytes) or ratio's between the circulating leukocytes?

• Were there differences in water/food consumption of the animals?

These data of the pre-treated alone should be presented at the beginning of the study.

4) It is unclear why the authors chose such an unusual model of experimental aspergillosis using a remarkably high inoculum of 3x10^8 conidia. Immunocompetent mice are typically resistant to infection with A. fumigatus even when using high doses. Could this somehow explain the partial loss of survival observed in control mice? The rationale for the use of such as aggressive model should be clearly explained.

5) Are the differences in lung CFUs sustained after 48 hr or do the surviving mice resolve the infection?

6) Is there a role for the P-selectin ligand interaction with its receptor in this model? This process is critical in leukocyte adhesion to the vessels and cooperative signaling between P-selectin and CXCR2 in neutrophils has been shown to increase adhesion and the effector functions of neutrophils in several lung injury models. This point should be at least addressed in the Discussion.

7) The fact that the functional impairments of neutrophils are also present in cells derived from the bone marrow implies that the alcohol treatment may have a direct effect on the neutrophil precursors in the bone marrow. Have the authors looked at metamyelocytes and promyelocytes in the bone marrow?

---

## [Author Response]

Essential revisions:1) The rationale to study aspergillosis in relation to chronic alcohol consumption is poorly justified in the paper. Indeed the authors cite the literature that ARDS is more common in chronic alcohol abuse and list a number of pathogens, but not Aspergillus. The rationale to study aspergillosis specifically remains unclear.

Thank you for addressing this Discussion. Aspergillosis is well known to be an important cause of morbidity and mortality to immunocompromised patients such as in hematological malignancies and transplants. However, there are several other risk factors that are not considered as traditional risks for developing aspergillosis. Kousha et al., 2011, published an extensive clinical review for pulmonary aspergillosis (PA). These authors discuss that chronic obstructive pulmonary disease (COPD) is an important risk factor for PA due to some reasons such as antibiotics, frequent hospitalization and comorbidities, such as alcoholism, among others. Moreover, PA can also affect patients who are mildly immunocompromised due to alcoholism, diabetes mellitus, chronic liver disease and low-dose corticosteroid therapy. Indeed, alcoholism is considered a risk factor for different clinical forms of aspergillosis since the decade of 1970 (Blum et al., 1978, Baddley, 2011 and Gustot et al., 2013). In fact, the rationale of this study is to understand the underlying mechanisms involved in increased susceptibility to Aspergillus pulmonary infection in chronic alcohol consumers. We also included some of that information in the Introduction section.

2) The use of an infection model with Streptococcal is used to validate the findings obtained with the aspergillosis model. However, it is not clear what is the actual relevance of adding this information, particularly in light of a few conflicting findings between the models, e.g., the levels of CXCL1 in the BAL. I would therefore recommend streamlining the manuscript to focus on the relevant findings on Aspergillus and removing the information on the Staphylococcus infection as it does not strengthen the main message of the manuscript.

Thank you for this suggestion. Indeed, our purpose in including the results with the S. pneumoniae model was to validate the mechanisms previously characterized for A. fumigatus infection. However, we also agree that this set of results do not strengthen the manuscript. We removed all data related to the S. pneumoniae model.

3) The in vivo experiments look very rigorous, however, additional information should be provided before infection to see how ethanol-fed mice are affected by the pre-treatment period alone. This is information can also be vital to mechanistically understand the observed neutrophil defects• Do the mice have systemic inflammation (elevated CRP/IL-1/TNF/CXCL1 levels)?

Thank you for this suggestion. Actually, we have performed a set of experiments to understand better the effects of alcohol consumption and collected the majority of the requested data. In case of systemic inflammation due ethanol consumption, we did not observe any increase in TNF-α, IL-1β and CXCL1 levels in serum from mice treated with ethanol, compared to control mice. We added this information in the manuscript, please see Figure 1.

• Can endotoxin be detected systemically?

We did not measure endotoxin in alcohol-treated mice. However, our results do not demonstrate a systemic inflammation that could indicate the presence of circulating endotoxins. Serum cytokines and chemokines levels, such as TNF-α, IL-1β and CXCL1 are in basal levels. TNF-α is one of the cytokines considered to play a pivotal role as a mediator of host’s response to LPS. Thus, whenever there is a presence of LPS, the release of systemic TNF-α is induced in high levels and can cause even lethality in mice or human subjects (Ruggiero et al., 1993 – Mediat of inflam 2:S43-S50; Wang et al. World J Gastroenterol 2002 Jun;8(3):531-6 and Hamilton, et al. – Scan J Infect Dis, 1992;24(3):361-8; Copeland et al., Clin Diagnot Lab Immunol, 2005 12:60-67). Moreover, LPS, together with ATP are also capable of induce inflammasome mediated IL-1β release and our model do not demonstrate this rise of IL-1β levels in serum (Nee LE et al., Kidney Int. 2004;66(4):1376-1386; Dinarello CA. Blood. 2011;117(14):3720-3732; Wang et al., Pharmazie – An Internat Jl of Pharmac Sci, 2014 69(9), pp. 680-684(5)).

• Do the mice lose bodyweight as a result of chronic alcohol consumption?

That is a very good question. We actually made a set of experiments prior Aspergillus infection and we collected data related to body weight change in mice weekly. In our work we did not observe significant changes in weight from ethanol-fed mice compared to control mice. We added this information in the manuscript, please see Figure 1.

• Are their differences in the absolute numbers of circulating immune cells (neutrophils/monocytes/lymphocytes) or ratio's between the circulating leukocytes?

Thank you for this question. Actually, in the first version of the manuscript we showed a graph with total cell count in mice after treatment. Now, we present a detailed graph with differential cell count in blood prior Aspergillus infection, please check Figure 1.

• Were there differences in water/food consumption of the animals?These data of the pre-treated alone should be presented at the beginning of the study.

Thank you for this comment. We actually collected data related to weekly food and liquid consumption in mice during ethanol treatment. Indeed, in our work we observed that the introduction of a diet based on ethanol consumption decreased the amount of food and liquid intake in mice, compared to control mice. However, the decreased consumption of food and liquids did not affect the overall weight of animals. We added this information in the manuscript, please see Figure 1.

4) It is unclear why the authors chose such an unusual model of experimental aspergillosis using a remarkably high inoculum of 3x10^8 conidia. Immunocompetent mice are typically resistant to infection with A. fumigatus even when using high doses. Could this somehow explain the partial loss of survival observed in control mice? The rationale for the use of such as aggressive model should be clearly explained.

Thank you so much for this comment. Indeed, we have addressed this question during the development of this work. In order to standardize inocula and extensively characterize the aspergillus model of pulmonary infection, our group previously demonstrated the infection profile in C57BL/6 mice in two different inocula: 1x10^8 and 3x10^8 conidia. In the lower inoculum, we observed lethality about 25% while in the higher inoculum was about 52% lethality. We attributed the increased lethality to the huge inflammatory response triggered by the high inoculum, since we have characterized the course of acute A. fumigatus lung infection in immunocompetent mice, investigating the immunological, pathological and tissue functional modifications (Malacco et al., 2020). This previous article paved the way for the choice of inoculum in the present work. We have also seen BALB/C mice mortality in immunocompetent model of Aspergillus infection with 1x10^8 inoculum (Malacco, et al., 2019. Front Cell Infect Microbiol Jan 11;8:453. doi: 10.3389/fcimb.2018.00453. eCollection 2018). Other works have used similar inocula and still showed different lethality rates in immunocompetent mice. Guerra et al. (PLoS Pathog. 2017 Jan; 13(1): e1006175) using BALB/C mice verified about 30% lethality with a 5x10^7 inoculum.

5) Are the differences in lung CFUs sustained after 48 hr or do the surviving mice resolve the infection?

Thank you so much this question. Again, we addressed some aspects of this question in another publication (Malacco et al., 2020). Using the same immunocompetent model for pulmonary aspergillosis, we demonstrated an intense neutrophil influx into the airways after 24 hours of infection, reaching a peak at 72 hours of infection. After that, neutrophils numbers progressively decreased until 168 hours of infection, when the complete spontaneous resolution of inflammation occurs, with a resolution index of 36 hours. Moreover, the high production profile of inflammatory mediators change after 3 days of infection with decrease of pro-inflammatory and up-regulation of anti-inflammatory cytokines and chemokines. These previous results suggest that, after 3 days post infection, lung CFU levels should decrease, at least in control (water-fed) animals. Unfortunately, we are not able to state if this is happening in ethanol-fed mice, nor if the differences between groups are sustained after that. However, taking into account our results: (i) lethality (Figure 2B); (ii) weight loss (Figure 2C) and (iii) inflammatory mediators (Figure 3A-F), we can assume that the inflammatory process is gradually diminishing in both groups (ethanol and water-fed mice), even with differences among them.

6) Is there a role for the P-selectin ligand interaction with its receptor in this model? This process is critical in leukocyte adhesion to the vessels and cooperative signaling between P-selectin and CXCR2 in neutrophils has been shown to increase adhesion and the effector functions of neutrophils in several lung injury models. This point should be at least addressed in the Discussion.

Thank you for your comment. Indeed, we agreed about the relevance of the selectin ligand and CXCR2 cooperation in neutrophil migration. The switch of β2 integrin from a low-affinity to a high-affinity state is an essential step for neutrophil crawling. We have added this point in the Discussion.

7) The fact that the functional impairments of neutrophils are also present in cells derived from the bone marrow implies that the alcohol treatment may have a direct effect on the neutrophil precursors in the bone marrow. Have the authors looked at metamyelocytes and promyelocytes in the bone marrow?

That is a very good question and represents one important point we are interested to explore. We already had collected samples of bone marrow and a hematologist analyzed them. Results showed no differences between neutrophils precursors in ethanol-fed mice compared to control mice. The results are presented as Supplementary file 1 and in the subsection “Chronic ethanol consumption did not induce systemic inflammation”.